# Low Host Abundance and High Temperature Determine Switching from Lytic to Lysogenic Cycles in Planktonic Microbial Communities in a Tropical Sea (Red Sea)

**DOI:** 10.3390/v12070761

**Published:** 2020-07-15

**Authors:** Ruba Abdulrahman Ashy, Susana Agustí

**Affiliations:** 1Red Sea Research Center, Division of Biological and Environmental Science and Engineering, King Abdullah University of Science and Technology, Thuwal 23955, Saudi Arabia; 2Department of Biological Sciences, Faculty of Science, University of Jeddah, Jeddah 23445, Saudi Arabia

**Keywords:** phage, marine bacteria, marine viruses, temperature, mitomycin C, viral production, lytic, lysogeny, Red Sea, eutrophic

## Abstract

The lytic and lysogenic life cycles of marine phages are influenced by environmental conditions such as solar radiation, temperature, and host abundance. Temperature can regulate phage infection, but its role is difficult to discern in oligotrophic waters where there is typically low host abundance and high temperatures. Here, we study the temporal variability of viral dynamics and the occurrence of lysogeny using mitomycin C in a eutrophic coastal lagoon in the oligotrophic Red Sea, which showed strong seasonality in terms of temperature (22.1–33.3 °C) and large phytoplankton blooms. Viral abundances ranged from 2.2 × 10^6^ to 1.5 × 10^7^ viruses mL^−1^ and were closely related to chlorophyll *a* (chl *a*) concentration. Observed high virus-to-bacterium ratio (VBR) (4–79; 16 ± 4 (SE)) suggests that phages exerted a tight control of their hosts as indicated by the significant decrease in bacterial abundance with increasing virus concentration. Heterotrophic bacterial abundance also showed a significant decrease with increasing temperature. However, viral abundance was not related to temperature changes and the interaction of water temperature, suggesting an indirect effect of temperature on decreased host abundance, which was observed at the end of the summertime. From the estimated burst size (BS), we observed lysogeny (undetectable to 29.1%) at low percentages of 5.0% ± 1.2 (SE) in half of the incubations with mitomycin C, while it increased to 23.9% ± 2.8 (SE) when the host abundance decreased. The results suggest that lytic phages predominate, switching to a moderate proportion of temperate phages when the host abundance reduces.

## 1. Introduction

Viruses are the most abundant organisms in the oceans, and high proportions are bacteriophages [1]. Thus, they exert significant control on bacterial mortality, responsible for killing 10 to 50% of the total concentration of bacteria in surface waters [2,3]. The number of marine viruses varies between 10^5^ viruses per mL in oligotrophic and deep-sea ecosystems [4,5,6], to 10^8^ viruses per mL in productive systems, with a total number of ~10^30^ virus particles estimated in the entire ocean [2,3]. Viral infections can be divided into two main categories: the lysogenic and the lytic cycles. In the lysogenic cycle, the phage DNA establishes stable interactions within the host cell, characterized as a temperate phage, and it allows the phage genome to integrate into the host cells until being induced by certain signals to become lytic [1,7]. In the lytic cycle, the virus reproduces, eventually leading to the destruction of the host cell, producing virions [7,8]. Both lytic and lysogeny cycles are reported to occur in marine bacteria, with losses of bacterial biomass due to viral lytic infections ranging from 20 to 50% [9]. Lysogeny promotes phage and host survival, especially under adverse conditions [8]. Viral infections decrease under exposure to ultraviolet (UV) radiation and other stressors [10,11], such as elevated temperature [12]. Additionally, temporal changes, such as low nutrients or low cell biomass, are usually associated with the lysogenic cycle [13]. The incidence of temperate phages is reported to differ depending on the trophic state of marine ecosystems. Lysogeny has been shown to be predominant in the oligotrophic waters of the Gulf of Mexico, whereas prophage induction was much lower in the highly productive waters of the Mississippi River plume [1,14,15]. Lysogeny in oligotrophic systems was identified as a survival strategy of viruses because of the low host abundance [16], although lysogeny was found more dominant in eutrophic estuarine waters than in oligotrophic waters [7,17], and temperate dynamics become increasingly significant at high host densities [18]. The occurrence of lysogeny was also found in warm oligotrophic waters, such as the Red Sea [19] and the Mediterranean Sea [20], although with strong seasonal variability.

It has been reported that temperature may also regulate the dynamics of infection and can differ among viruses that infect the same host. Although viruses are dependent on their hosts for replication, they may be more tolerant than their hosts to thermal stress, determining that the temperature distribution of the virus-host system is set by the host [21]. Most viruses are capable of surviving at low temperatures, whereas high temperatures may result in the loss of viral infectivity, leading to an inactivation of virus particles [22,23,24,25]. In Tampa Bay, Florida, the occurrence of lysogenic bacteria was found to have a significant proportion of bacterial population during certain times of the year, which contrariwise reflected temperature, host abundance, and bacterial and primary production [14]. The prevalence of lysogeny was found in low water temperature environments, including North Atlantic Oceans [26], polar, mesopelagic, and deep-sea waters or periods of low host abundances [14,15,27,28,29].

The effect of temperature on viral infection can modify the kinetics of viral lysis, which in turn develop viral resistance [22,30,31] or influence the switch from lysogenic to lytic cycles [32]. However, it is still far from understood whether the shift from lysogeny to lytic viral replication that is regulated by temperature illustrates a global pattern or is related to local processes [29]. Environments that are comprised of mutagenic pollutants may have a frequent occurrence of prophage induction, including a sudden temperature change that could be an essential inducing agent for natural lysogenic viral production in marine environments [17]. A significant positive effect of temperature was found on the total abundance of viruses in the coastal waters of northeastern Taiwan [33]. Bacteriophage 9A was isolated from particle-rich Arctic seawater and was found stable at low temperatures and extremely thermolabile at 25 °C and above [34]. However, increases in temperature and rates of cell growth have been identified as inducing the lytic cycle [17].

The Red Sea is one of the warmest seas on earth [35], is a semi-enclosed sea experiencing warming rates exceeding global ocean warming [36], and is considered to be mostly oligotrophic [35,37]. Temperatures vary seasonally between 22 and 32 °C [35]. Some areas of the Red Sea have higher nutrient concentrations, as reported for some shallow coastal lagoons [38], or the Southern Red Sea, which receives nutrient inputs from the Indian Ocean through the Gulf of Aden [39]. Temperature variation might affect the virus and host dynamics in the Red Sea [36]. However, the role of water temperature on viral dynamics and/or in the shift of viral life strategies in the Red Sea is still unexplored. Although several studies have been performed on microbial communities and bacteria [40,41,42], studies on viral dynamics in the Red Sea are still limited to communities in coral reefs [43,44] and in the deep-sea brines [45]. A previous study in oligotrophic Red Sea coastal waters showed high viral dynamics, with more than 50% of inducible lysogenic bacteria occurring during the wintertime when host abundance was low [19], suggesting that lysogeny may be an important mechanism for viral replication.

Here, we aim to quantify the planktonic viral abundance and lytic-lysogeny seasonal dynamics in a coastal lagoon of the central Red Sea. Temperature variability in the study area exceeds the maximum reported for the Red Sea, and nutrient dynamics allow for the formation of phytoplankton blooms [38,46]. Our goal is to discern whether temperature changes and host abundance determine changes in the proportion of lysogenic bacteria and to identify the influence of temperature and host abundance on viral infections.

## 2. Materials and Methods

### 2.1. Study Sites and Sample Collection

Time-series sampling was conducted biweekly from October 2017 to October 2018 at a coastal lagoon in the central Red Sea of Thuwal, Saudi Arabia, at 22.39° N, 39.14° E (Figure 1). Seawater samples were collected from sea surface water (1 m) in a 5 L polypropylene container, presterilized with 4% of HCl, and prerinsed with the same collected seawater on the same day of collection for analysis of microbial communities, viral production (VP), and lysogeny proportion. The collected seawater samples were transported to the lab, and duplicated subsamples (1.5 mL each) were taken for the enumeration of natural viral and heterotrophic bacterial abundances by flow cytometry (FCM-the BD FACSCanto™ II, © 2006, Becton, Dickinson and Company). The remaining water was prefiltered (20 µm and 2 µm, respectively) and was used for virus-free water (VFW) preparation and the different incubation experiments.

### 2.2. Environmental Parameters Measurements

Sea surface water temperature (°C) and salinity (PSU) were measured using an Ocean Seven 305 Plus CTD device (Idronaut, Brugherio, Italy) for 5–15 min. Chlorophyll *a* (Chl *a*) concentrations were analyzed as described by [46], by filtering 300 mL of seawater through a Whatman glass microfiber GF/F filter (Sigma-Aldrich, Taufkirchen, Germany). The filtered water was extracted in 90% acetone and estimated with the nonacidification technique using a Turner Design Trilogy Fluorometer.

### 2.3. Virus Reduction

The virus reduction method [47,48,49] was performed to measure VP rates and lysogenized bacterial cells, as described by [19]. The collected prefiltered seawater samples were first filtered through a 0.2 µm-pore-size membrane (described below) and then were ultrafiltrated through an ultrafiltration cartridge with a 30,000 Daltons molecular-weight cutoff to reduce viral particles and obtain 500–600 mL of VFW [48,50].

### 2.4. Incubations for Measurements of Lytic Viral Production and Lysogeny Induction

Three liters from each of the prefiltered seawater samples (20 µm and 2 µm) were concentrated to obtain a volume of 50–70 mL using an ultrafiltration system (Amicon^®^ Stirred Ultrafiltration Cell 8050 Millipore 50 mL-Merck Millipore) with a 0.2 µm-pore polycarbonate filter to maintain heterotrophic bacterial cells and reduce virus concentrations [50]. The concentrated heterotrophic bacterial cells were rinsed five to six times with 100 mL of VFW and then resuspended until a volume of 250–300 mL was reached [50,51,52]. The Amicon^®^ stirred ultrafiltration cell is an advantageous concentration system as it gives high flow rates and is capable of concentrating samples rapidly but gently, as was successfully used previously in quantifying living cells [53,54]. The system uses magnetic stirring and pressure-driving filtration, resulting in a short time required for cell concentration (i.e., less than half an hour in our samples). After the dilution of heterotrophic bacterial cells with the VFW, the recovery percentage of the concentrated heterotrophic bacteria was high and calculated as 70% ± 0.08 (mean ± SE) of the initial concentration.

The washed heterotrophic bacterial concentrates with VFW were divided into six acid-rinsed glass flasks with a volume of 40–50 mL and were used for the incubation experiments. For inducing the lysogenic phase, three replicates were treated with 1.0 µg mL^−1^ of the chemical treatment, mitomycin C (Sigma Chemical Co. catalog no. M-0503; 1 mg mL^−1^ stock solution, dissolved in deionized water [DI]) [50,55], and the other three replicates were kept as untreated controls for the quantification of viral and heterotrophic bacterial abundances compared with mitomycin C-treated samples. Mitomycin C is an effective inducing agent for several marine ecosystems that is not found naturally in marine environments [56]. Mitomycin C is a chemical compound that damages the DNA of the host cell and activates a DNA repair mechanism such as the RecA protein, which cleaves a repressor to induce prophages, thereby converting lysogenic cycles to lytic cycles [50]. The flasks were then incubated for 24 h in the dark [47,55,57] at the same in situ water temperature measured at the station at the time of sampling, which varied across the study between seasons from ~21 to 34 °C. From time 0 to 24 h (T_0_ to T_9_), 2 mL was taken every three hours from each flask, and preserved using 80 µL of 25% glutaraldehyde [58] and stored at −80 °C until enumeration of the viruses and heterotrophic bacteria. For overnight sampling, a fraction collector, Gilson, Inc-FC204 with multiple heads and peristaltic pumps, was used for the programmed automatic sampling. Eight cryovials (2 mL each) with 80 µL of 25% glutaraldehyde were placed (without lids) in the designed racks surrounded by dry ice to preserve the samples in cold conditions. The fraction collector was monitored to transfer the samples from the flasks to the cryovials every three hours automatically. The effectiveness of fixing and storing the samples automatically compared to manually fixing was tested previously by [19] and no significant differences were found.

### 2.5. Viral Production Rates

The virus reduction method was prepared as described above to estimate lytic viral production (LVP) rates [47]. LVP rates were corrected for possible heterotrophic bacterial loss or increase during the concentration process by dividing the in situ bacterial concentrations by the initial bacterial abundances in both VP measurements. LVP rates were estimated following [59] from the slope of the relationship between the minimum and the maximum viral abundance versus time for the first six to 12 h of incubation in order to exclude viruses produced by new infections during the incubation.

### 2.6. Flow Cytometry

Subsamples for enumeration of viral and heterotrophic bacterial populations were analyzed using a BD FACSCanto™ II flow cytometer (© 2006, Becton, Dickinson and Company). Currently, flow cytometry (FCM) is a well-established, relatively straightforward [60], and high-throughput technique [61] in estimating the abundances of aquatic microbes, including viruses and bacteria [60,61]. A series of simplified protocols to identify and enumerate virus populations were carried out as described by [19] following the protocols of [60,61], with some modifications such as the following: The samples were removed from storage (−80 °C) and kept melting at room temperature. Sample preparation for viral abundance was carried out under sterilization conditions. In the beginning, a water bath was turned on at 80 °C. Control samples (a blank that was subtracted from all counts) were prepared by adding 50 µL of an autoclaved and prefiltered 0.2 µm seawater sample and diluted into 475 µL of Tris-EDTA buffer (TE, 10 mM Tris and 1 mM EDTA, pH = 8) in order to obtain an event rate ranging from 300 to 700 events s^−1^ [58], with an addition of 5 µL of SYBR Green I (1:2000). Then, the samples were prepared using the same preparation steps for the control samples but with the addition of 50 µL of the time series samples instead of the autoclaved seawater. The tubes were then incubated for 10 min in the 80 °C water bath and were allowed to cool down for 5 min in the dark at room temperature.

The flow rate was measured before and after running in the FCM using 1 mL of autoclaved Milli-Q that included 10 µL from a total concentration of 10^4^ fluorescent beads. An adequate collection time of 60 seconds was used at the low flow rate speed. The threshold of FCM for determining and identifying virus populations was applied for green fluorescence (GF) versus side scatter (SSC) and versus red fluorescence (RF). FCM analysis differentiated three viral subpopulations with different fluorescence properties, as described by [61]. The virus subpopulations that we distinguished are as follows: V1, with the lowest green fluorescence, V2, with the midlevel fluorescence, and V3, with the high fluorescence. Heterotrophic bacteria were analyzed and prepared following the protocol of [60] by adding 400 µL of each sample stained with 4 µL of SYBR Green I (1:1000), and then the samples were kept in the dark for 10–20 min before running in the FCM. The BD FACSCanto™ II threshold was set up as the RF versus the GF. Data were recorded and saved in the BD FACSCanto™ II and then were analyzed by quantifying the events of viral and heterotrophic bacterial populations using FlowJo (Version 10.1-Tree star. Inc, USA) as [62]. For data analysis, refer to methods section (statistical analysis).

### 2.7. Burst Size and the Percentage of Lysogenic Bacteria

Control samples of natural seawater (no virus reduction) were incubated in parallel to the viral reduction incubations for the calculation of the burst size (BS) (i.e., the number of phages produced per infected bacterium). The BS was assessed from viruses produced during incubations. BS was then calculated by subtracting the produced virus (VP*_c_*) in the untreated control samples from the number of produced virus in the virus-reduced samples (VP*_r_*), which demonstrates the net increase in the number of phages that were released from infected bacteria, and then dividing by the number of bacteria killed (B*_dead_*) during the incubations by infection [63], as follows:(1)$$BS=\left(VP_{c}-VP_{r}\right)/\;\left(B_{dead}\right)$$

Lysogenic bacterial percentage (%) was calculated following [50]:(2)(Vmc−Vc) /BS/BA×100
where V*_mc_* represents the maximum viral abundance in the mitomycin C-treated samples during incubation, and V*_c_* refers to the maximum viral abundance in the control untreated samples. BS applies to the burst size, and BA is related to bacterial abundance at the onset of each incubation experiment.

### 2.8. Statistical Analysis

JMP PRO 14 software (JMP^®^, Version *< 14 >* SAS Institute Inc., Cary, NC, 1989–2019) was used for statistical analysis. The bivariate test was applied to FCM normalized data to determine linear regression between viral and heterotrophic bacterial abundances, and environmental conditions, and the significance (*p*-value) of this correlation. Correlation between variables was tested with a multivariate analysis using a nonparametric Spearman’s ρ correlation test. For the *r*-ratio, values in the range of −0.5 < *r* < 0.5 were considered weakly correlated, while values in the range of *r* ≤ −0.5 and *r* ≥ 0.5 were considered strongly correlated. In addition to these thresholds, *r* = 0 indicated no correlation, *r* < 0 is a negative correlation, and *r* > 0 is a positive correlation. For *p*-values, the significance threshold was ≤ 0.05. Multiple linear regression test was used to learn more about the interactions between the several environmental variables with phage infections and heterotrophic bacterial abundance.

## 3. Results

### 3.1. Environmental Parameters Measurements and Microbial Abundances

Mean monthly records of water temperature and chl *a* parameters, in addition to viral and heterotrophic bacterial abundances together with virus-to-bacterium ratio (VBR), are presented in Figure 2. Sea surface water temperature during the sampling period ranged from 22.1 to 33.3 °C (28.7 ± 1.0 °C, mean ± SE) with the minimum value detected in January and the maximum in September (Figure 2). Chl *a* concentrations ranged from 0.5 to 5.1 µg L^−1^ (1.5 ± 0.4 µg L^−1^, mean ± SE), showing a large bloom in October 2018 with the lowest value in February (Figure 2A). We determined the three subpopulations of viruses; however, the low fluorescence viruses (V1) dominated the viral community along with the study with averaged percentages as V1: 76.7% ± 3.7, V2: 19.7% ± 3.1, and V3: 3.6% ± 0.6 (mean ± SE). Viral abundances ranged from 2.2 × 10^6^ to 1.5 × 10^7^ cells mL^−1^ (5.7 × 10^6^ ± 9.8 × 10^5^ cells mL^−1^, mean ± SE), showing an increase in September, with a large peak at the beginning of October 2018 and lower values during the winter and the spring (Figure 2B). Abundances of heterotrophic bacterial cells ranged from 2.2 to 6.7 × 10^5^ cells mL^−1^ (4.7 × 10^5^ ± 3.6 × 10^4^ cells mL^−1^, mean ± SE), as the peaks were observed in October and November 2017, with increased values in the spring and the lowest in September (Figure 2C). From viral and heterotrophic bacterial abundances, we calculated VBR as they ranged from relatively low to high values (Figure 2D). The lowest VBR values were observed in spring 2018 and the highest in fall 2018 (Table 1).

### 3.2. Correlations between Environmental Parameters and Microbial Communities

The nonparametric Spearman’s rank correlation test was used to determine the correlation between all the environmental parameters and viral and bacterial abundances (Figure 3). According to the monthly records of the environmental parameters, a positive and significant correlation was suggested only between salinity and water temperature (ρ = + 0.7214, *p* ≤ 0.0001); however, the relationships between the other environmental variables were all insignificant. For microbial communities, viral abundance had a positive and significant correlation with chl *a* concentration (ρ = + 0.6073, *p* = 0.0008); and bacterial abundances with water temperature were shown to have a significant negative relationship (ρ = − 0.4348, *p* = 0.0234). Additionally, the correlation between the percentage of lysogenic bacteria and chl *a* showed positive and significant relationship (ρ = + 0.6009, *p* = 0.0388) (Figure 3).

The relationship between microbial communities with respect to environmental parameters, including water temperature, chl *a* concentration, and salinity was also detected using linear regression analysis. There was a significantly positive linear relationship between viral abundance and chl *a* concentration (R^2^ = 0.4578, *p* < 0.0001) (Figure 4A), while it was negatively correlated with heterotrophic bacterial abundance but significant (R^2^ = 0.1841, *p* = 0.0255) (Figure 4B). Heterotrophic bacterial abundance had a significant negative relationship with water temperature (R^2^ = − 0.1783, *p* = 0.0282) (Figure 4C). Additionally, the linear regression analysis between the percentage of the low fluorescence subpopulation viruses (V1) and the total viral abundance determined a significant positive relationship (R^2^ = 0.5666, *p* < 0.0001), indicating that V1 increased to dominate viral community with the increase in the total virus abundance. However, there were no significant relationships between viral abundance and water temperature (R^2^ = 0.0578, *p* = 0.2271, Figure 4D), and between heterotrophic bacterial numbers and chl *a* concentration (R^2^ = 0.0853, *p* = 0.1394, Figure 4E). We found strong positive and negative significant relationships between VBR with viral and heterotrophic bacterial abundances, respectively (R^2^ = 0.8313, *p* = 0.0001); (R^2^ = − 0.5764, *p* = 0.0001). VBR was positively significantly correlated to chl *a* concentration (R^2^ = 0.3849, *p* = 0.0006) and salinity (R^2^ = 0.1535, *p* = 0.0432).

The multiple linear regression analysis of the interactions among heterotrophic bacterial abundance, viral abundance, and the environmental parameters indicated that there is a strong negative significant relationship between viral and bacterial abundances and between bacterial abundances and viruses with chl *a* (Table 2). Although chl *a* concentration showed a positive relationship with bacterial numbers, the relationship between the two variables was weakly significant (Table 2). The finding that the abundance of heterotrophic bacteria was not significantly related to water temperature (Table 2) may suggest that the significant negative correlation observed (Figure 3 and Figure 4) is an indirect effect of temperature on bacterial numbers.

### 3.3. Viral Production and the Percentage of Lysogeny

After viral reduction and during the 24 h incubation experiments, variable changes were observed in viral abundances in both mitomycin C-treated and untreated control samples, which can be expected as a result of inducible lysogenic bacteria (Appendix A). The increase in viral abundance in the mitomycin C-treated samples was higher than in the controls for the incubations made in December, March, June, August, and October, indicating the induction of prophages. A peak in prophage induction was observed after three to six hours for most of the incubations (Appendix A). The estimated average BS (*n* = 7) was ~15 ± 5.3 (mean of the new phages per lysed bacteria) (Table 1). The BS was not estimated for each incubation experiment, and thus we could not obtain it for all the study period. The percentage of lysogenic bacteria varied from undetectable in the winter, in the spring, and in July, with low percentages occurring in March, August, and September, to higher percentages observed in the fall of 2017 (beginning of December) and June 2018, with the maximum of 29.1% detected in October 2018 (Table 1, Figure 5 and Figure 6). VP rates were estimated from the initial slope of viral abundances of the untreated controls for each incubation experiment in order to predict lytic infections in the time of lysogeny absence. LVP rates (Table 1 and Figure 6) varied highly during the study from negative values to a maximum of two peaks detected in April and September with 5.8 × 10^5^ and 5.6 × 10^5^ viruses mL^−1^ h^−1^, respectively. We found negative correlations between the percentage of lysogenic bacteria and bacterial abundances (R^2^ = 0.0876, *p* > 0.05) and between the percentage of lysogenic bacteria and the LVP rates (R^2^ = 0.1073, *p* > 0.05).

## 4. Discussion

The big trophic changes of the Red Sea lagoon under study highlights the significant role of trophic conditions and high temperature on heterotrophic bacterial dynamics, where periods of high water temperature influenced the lysogenic cycle by decreasing the abundance of host cells. Our results show that lytic phages dominate prophage production in the warm coastal waters of the study area over time. Moreover, the results suggest that the induction of lysogenic bacteria occurred primarily during times of decreased host abundance. We identified prophage induction in almost half of the incubation experiments performed in the coastal Red Sea, although percentages were not high, with a maximum of 29.1% of lysogenic bacteria observed in October when bacterial abundance was relatively low (2.7 × 10^5^ cells mL^−1^) and water temperature was high (>32 °C). In contrast, we detected the lowest lysogeny percentage in March at 2.8% when bacteria reached higher abundance (6.0 × 10^5^ cells mL^−1^) and water temperature was relatively low (25.8 °C). Our results confirmed other previous observations, which supported our goals, indicating the indirect effect of temperature on host abundance, as shown by the significantly negative relationship between bacterial abundance and water temperature.

The high VBR found here agrees with the predominance of lytic replication phase. The VBR values were similar to those reported for eutrophic waters [4], although there was a large variation in VBR corresponding to the strong temporal variability observed in the lagoon. High VBR values could also be influenced by the presence of benthic communities (seagrass meadows) in the lagoon, as well as big seasonal changes in trophic conditions, as the system changed from oligotrophic to periods of phytoplankton blooms during the fall [46]. Chl *a* concentrations revealed high levels ranging from 0.5 to 5.1 µg L^−1^ in the lagoon, higher than those reported for surrounding coastal waters (0.1 to 0.6 µg L^−1^, [46]). We found a significant positive correlation between viruses and chl *a* concentrations, indicating a substantial influence from the phytoplankton blooms. This is in contrast with the results of [19] for an oligotrophic coastal Red Sea area and other oligotrophic areas such as the NW Mediterranean [20], where viral abundances were not significantly correlated with chl *a* and other measured environmental parameters. The peaks in viral abundance did not correspond with bacterial peaks, as was observed in the NW Mediterranean Sea [20] and in the oligotrophic Red Sea open waters [19], but they were influenced by the phytoplankton blooms.

Seasonal studies on viral abundance in Tampa Bay, Florida [64] and other eutrophic subtropical environments [57] have suggested that there is a probable correlation between viral concentration and water temperature as indicated by the maximum virus concentrations during spring and summer months and the minimum in the winter. Nevertheless, in our seasonal study, viral abundances were not positively correlated with water temperature; however, the number of viruses reached a peak in the fall when water temperatures were still high. Viral abundance in this study ranged widely, from 10^6^ viruses per mL, similar to those found in oligotrophic Red Sea areas [19], to 10^7^ viruses per mL, as reported for eutrophic coastal waters elsewhere [33,65,66]. As observed in other studies [26,61], the viral community here was dominated by the lower fluorescent bacteriophages (V1), which associated with bacterial abundances [26]. We found that bacterial abundances were significantly and negatively correlated with water temperature and the lowest abundance was observed at the end of September 2018 (late summer) when water temperature reached a maximum of 33.3 °C. The highest bacterial abundance was, however, recorded in fall 2017 (October) when water temperature decreased to 30 °C and the phytoplankton bloom receded. Bacterial numbers were lower when compared to other coastal eutrophic waters [66] but were similar to bacterial numbers in the Red Sea coastal harbor [42].

Over twelve months, we used the viral reduction method and the chemical inducing agent, mitomycin C [50,64,67], to examine lysogenic proportions. The difference in VP between mitomycin C-treated and untreated control samples was attributed to the induction of lysogenic bacteria [50,68]. Prophage induction often occurred between three to 12 h of incubation after adding (1 µg mL^−1^) of mitomycin C, showing earlier induction occurrence than in the previous study in the Red Sea [19], where prophage induction resumed after 18 h of incubation. This is similar to [68], which demonstrated that in mitomycin C-treated samples, total viral abundances had the largest increase between six and 12 h of incubation relative to untreated controls. The BS in this study was estimated at 15 ± 5.3 viruses per bacterium, which was similar to the BS estimated in the oligotrophic coastal Red Sea [19]. These values are also similar to those reported for the oligotrophic ocean and the Gulf of Mexico (15 to 54 [50]; and 18.9 [69], respectively), 12 for the Sargasso Sea and North Atlantic [70], and an average BS of 19.8 from various estimates from other oligotrophic marine environments [71].

Although lysogeny is believed to be more prevalent in eutrophic estuarine waters than in oligotrophic waters [7,17], our results show that lysogenic phage production was less dominant in a eutrophic coastal lagoon than in Red Sea oligotrophic waters [19]. This study showed the percentage of lysogenic bacteria was low, representing on average, 7.2% ± 2.9 (SE) and ranging from undetectable to 29.1% although lysogeny in oligotrophic Red Sea waters reached 50% [19]. Those results agree with those of [14,50], which found a higher percentage of lysogeny in oligotrophic waters than in coastal eutrophic waters of the Gulf of Mexico, as was also shown by [72] for the coast of California. In contrast, the study of [17] reported that inducible prophage was found in 43% of marine heterotrophic bacterial isolates, indicating that lysogens contained a significant proportion of the heterotrophic microbial population. Likewise, Weinbauer et al. [28] detected that up to 84% of bacteria in the Mediterranean and Baltic Seas were induced, with the highest percentages found in deeper waters (800–2000 m). Despite the persistent phytoplankton blooms in the lagoon area, LVP rates in this study showed moderate values averaging 2.6 × 10^5^ viruses mL^−1^ h^−1^ (±6.0 × 10^4^ SE), similar to those reported for oligotrophic coastal Red Sea waters [19]. The values were comparable to those reported by [70] for the oligotrophic Sargasso Sea and showed lower rates than those reported for temperate coastal waters [47,73,74], where VP rates ranged from 10^6^ to 10^7^ viruses mL^−1^ h^−1^. Based on the statistical analysis results of our study, the lytic viral infections were undetectable during the occurrence of the maximum proportion of lysogenic bacteria, suggesting the decaying of viral abundances.

Water temperature plays an essential role in microbial growth regulation, and therefore it may affect both viral lifecycles and production [75,76]. In our study, the final model of multiple regression analysis test showed a nonsignificant relationship between heterotrophic bacterial abundance and water temperature; however, the significant results from the other statistical analysis tests used suggest that the effect of water temperature on prophage induction in the Red Sea was indirect, by negatively affecting the host. The study of [18] suggested that the dynamics of temperate phages become increasingly significant at high host densities, where we observed a small percentage of lysogenic bacteria. Nonetheless, during the periods of declining host abundance, lysogeny becomes a more relevant strategy for the viral population. Our findings are consistent with other studies reporting that lysogeny in cyanobacteria was primarily correlated with ambient host abundance [14,15,27].

In our study, during August and September when the water temperatures were high, we recorded an increase in lysogeny associated with a decrease in bacterial abundance, suggesting some temperate phage integrate into the host genome. This suggests that they are capable of being stable until environmental conditions improve. As suggested by several authors [14,15,27,29,77], lysogeny predominance may occur either in low-temperature marine environments, such as polar, mesopelagic, and deep-sea waters or during low host abundance. However, Lara et al. [6] found a predominance of lytic phages over the lysogenic cycles in the bathypelagic ocean. Our results show that for warm tropical waters, lytic phages predominate, switching to a moderate proportion of temperate phages when the host abundance decreases. The big trophic changes within the lagoon enabled us to identify the role of trophic conditions and high temperature on bacterial dynamics, where periods of lysogeny under high water temperature conditions were induced by reducing host abundance. Declines in bacterial numbers during the summer may be related to the consistent decrease in phytoplankton and the associated reduction of organic carbon available for bacteria, although other aspects such as increased protist grazing [20,40] or UV radiation damage [78] can also reduce bacterial abundances.

Lysogeny can also benefit the host by improving the host’s resistance to stressors through the expression of advantageous genes carried by the virus [14,68]; thus, the host may be provided protection by the virus from stressors, such as UV radiation [8]. Considering the high variability of conditions in nature, the conditions that are controlling the switch from the lysogenic to the lytic phase need to be identified and understood, yet they remain mostly unknown [8,14]. Another aspect that is poorly addressed is the analysis of the relative contribution of archaea and heterotrophic bacteria to the observed viral production [79]. The methodology we used does not allow discrimination between archaea and heterotrophic bacteria and their viruses, where recent studies indicate that marine archaea and archaeal viruses are active and relevant components of marine microbial assemblages [79].

## 5. Conclusions

In summary, our results indicate that, in a coastal lagoon in the tropical Red Sea, lytic cycles dominate the infection of bacteria and the associated phages exert a tight control of the bacterial population in waters studied. Our findings suggest that high summer temperatures and the associated reduction in bacterial population indirectly induced lysogeny, although lysogeny did not represent a high proportion of total phage infection. Our study contributes to understanding the changes in viral abundance and lysogeny, which are highly dynamic in the warm and saline waters of the Red Sea and helps to discern the role of host and temperature in the switching of viral phases. However, further studies are necessary to determine natural environmental factors that could increase the impact of lysogeny in controlling the abundance and genetic diversity of marine microbial communities.

## Figures and Tables

**Figure 1 viruses-12-00761-f001:**
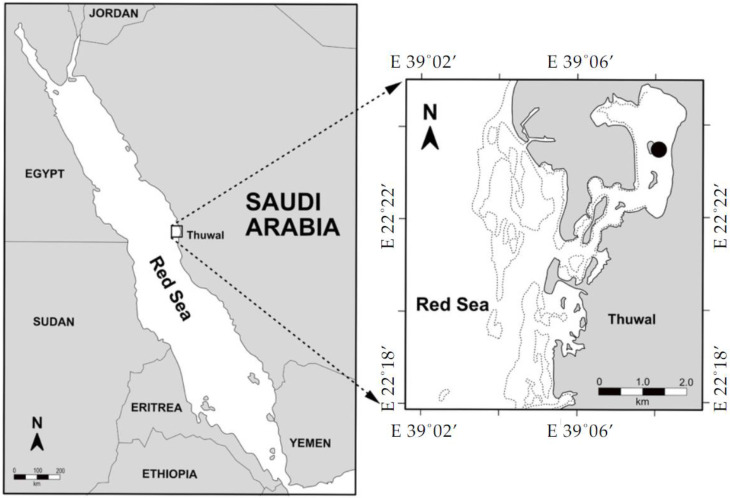
Location of the sampling station in the coastal central Red Sea (Saudi Arabia).

**Figure 2 viruses-12-00761-f002:**
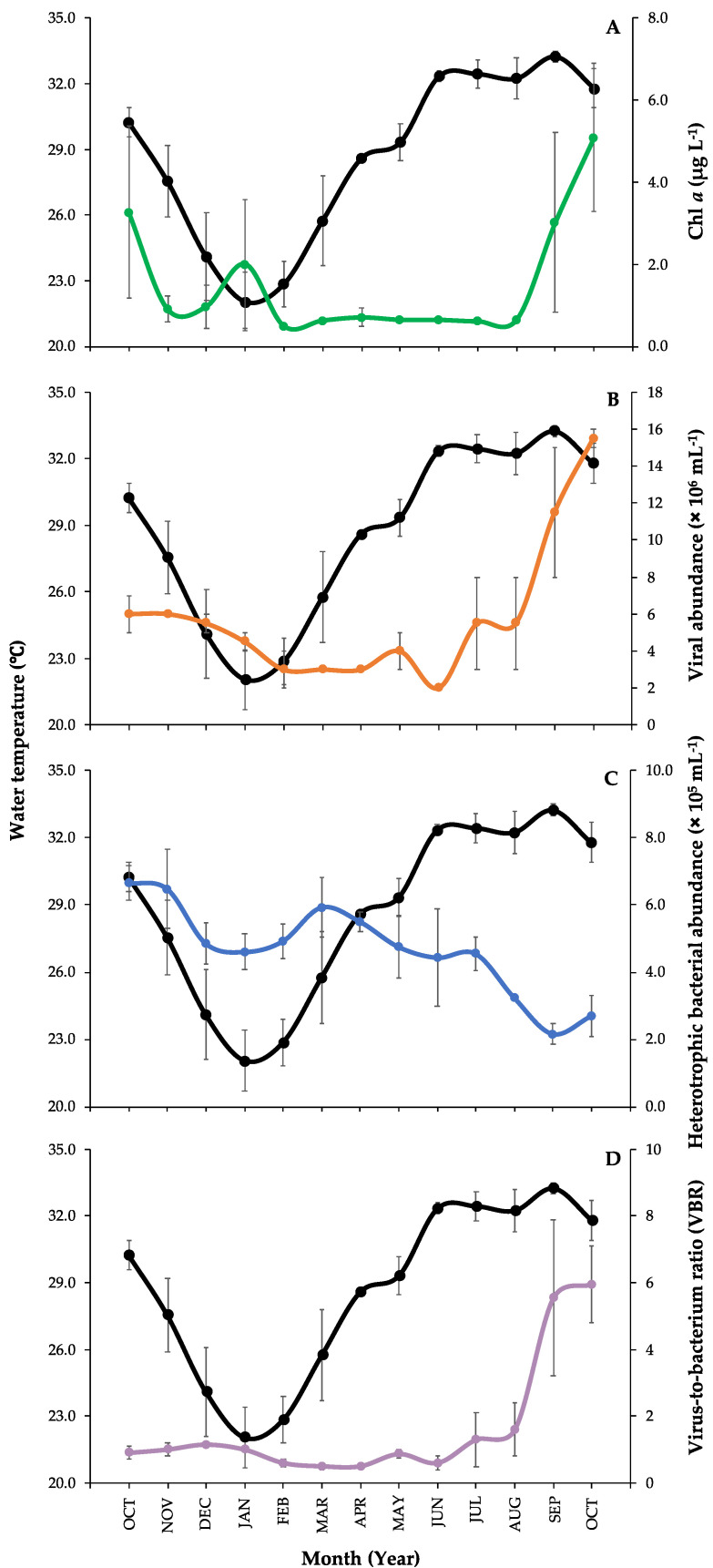
Temporal variability (monthly average ± SE) on different environmental parameters studied during the time series sampling in the coastal lagoon in the Red Sea (2017 to 2018). (**A**) Chl *a* concentration (green line and dots), (**B**) Viral abundance (orange line and dots), (**C**) Heterotrophic bacterial abundance (blue line and dots), and (**D**) Virus-to-bacterium ratio (VBR) (purple line and dots). The black line in all the plots corresponds to the surface water temperature of the Red Sea coastal lagoon area.

**Figure 3 viruses-12-00761-f003:**
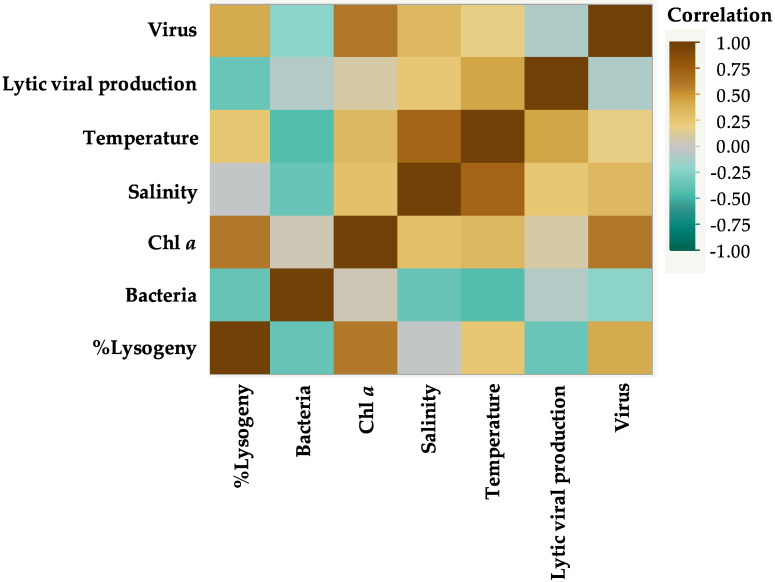
Correlation matrix between all the environmental parameters, viral and bacterial abundances, the percentage of lysogenic bacteria (%), and lytic viral production (LVP) rates (mL^−1^ h^−1^) observed during the study.

**Figure 4 viruses-12-00761-f004:**
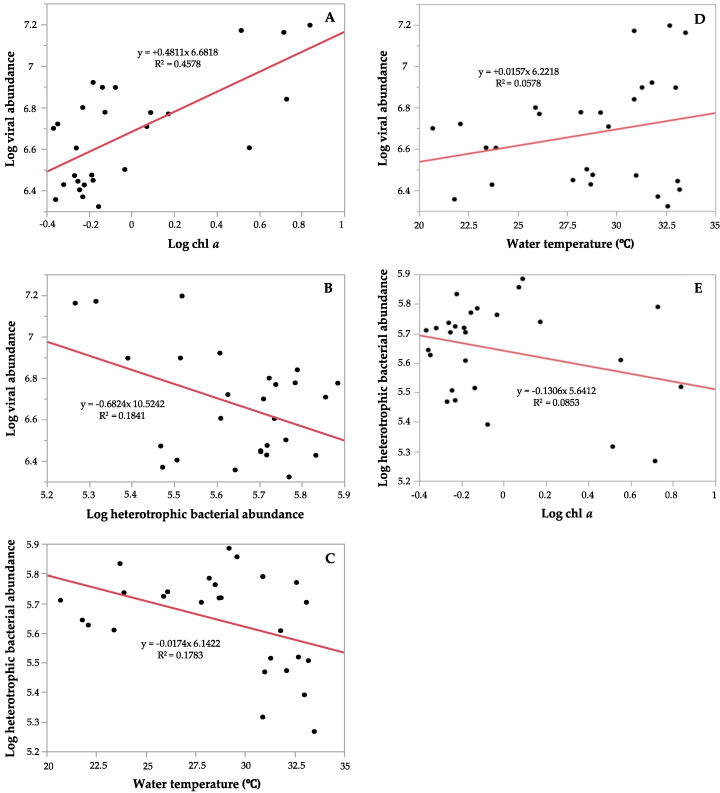
The significant relationships observed between viral abundance with (**A**) Log chl *a* concentration (*p* = 0.0001), and with (**B**) Log heterotrophic bacterial abundance (*p* = 0.0255); and between (**C**) Log heterotrophic bacterial abundance with water temperature (*p* = 0.0282). No significant relationships between (**D**) Log viral abundance with water temperature (*p* = 0.2271), and between (**E**) Log heterotrophic bacterial abundance with log chl *a* concentration (*p* = 0.1394). The red line fits the linear regression.

**Figure 5 viruses-12-00761-f005:**
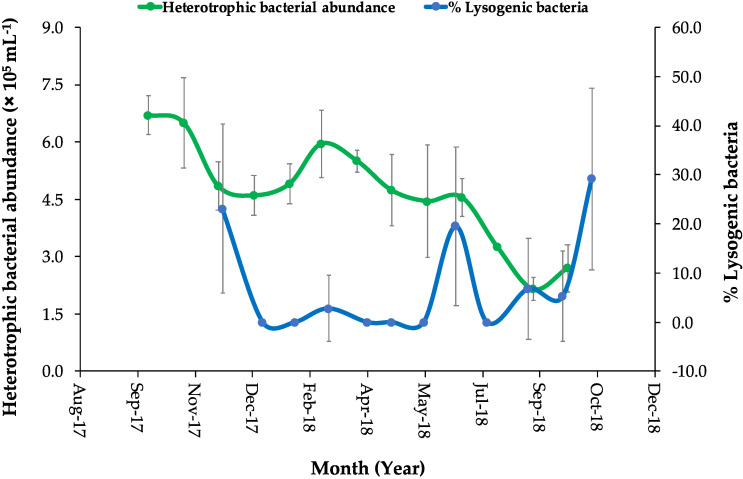
Percentages of lysogenic bacteria (blue line and dots) and bacterial abundance (green line and dots) quantified in the natural microbial communities of the coastal Red Sea lagoon (December 2017 to October 2018). The vertical bars encompassed the error bars. The error bars of the bacterial abundances were generated from two replicates. The error bars of lysogenic bacterial percentage were calculated for only the detectable lysogeny proportion.

**Figure 6 viruses-12-00761-f006:**
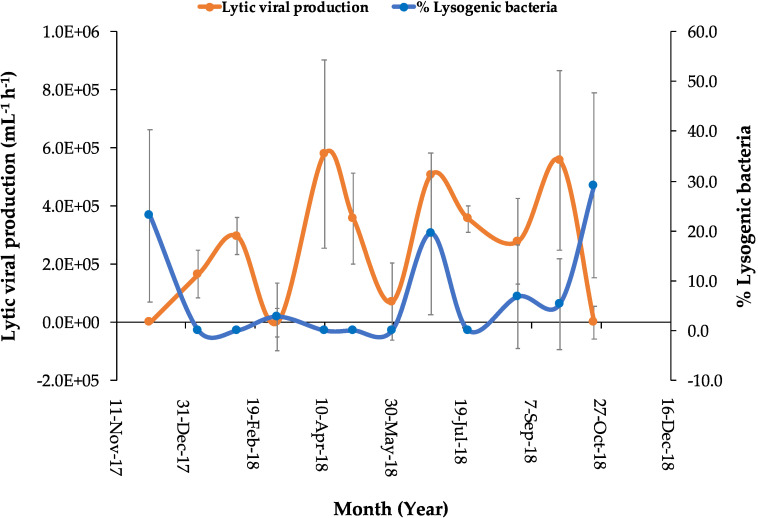
Lytic viral production (LVP) (orange line and dots) and percentages of lysogenic bacteria (blue line and dots) during the incubation experiments from December 2017 to October 2018. The vertical bars encompassed the error bars.

**Table 1 viruses-12-00761-t001:** Average (±SE) with the minimum and the maximum values of lytic virus production (LVP) obtained from viral reduction, estimation of lysogenic bacterial percentages based on induction by mitomycin C, burst size (BS) for monthly individual incubation experiments since December 2017 to October 2018, and virus-to-bacterium ratio (VBR) from viral and heterotrophic bacterial abundances during the time series sampling from October 2017 to October 2018.

Lytic Virus Production (VP) (mL^−1^ h^−1^)		Lysogeny (%)		Burst Size (BS)		VBR (Ratio)	
**Mean ±** **SE**	**(MIN/** **MAX)**	**Mean ± SE**	**(MIN/** **MAX)**	**Mean ± SE**	**(MIN/MAX)**	**Mean ± SE**	**(MIN/** **MAX)**
2.6 × 10^5^ ± 6 × 10^4^	N/D–5.8 × 10^5^	7.2 ± 2.9	N/D–29.1	15 ± 5.3	2.4–44.9	16 ± 4	4–79

N/D = Not detected.

**Table 2 viruses-12-00761-t002:** Multiple linear regression test model showing the interactions among the environmental parameters (Water temperature and chl *a* concentration) and viral abundance with heterotrophic bacterial abundance.

Independent Variables and Viral Abundance	Heterotrophic Bacterial Abundance		
Interactions/Units	Parameters Estimates ± SE	*t*-ratio	*p*-value
Intercept	8.3505 ± 1.0653	7.84	<0.0001
Virus/log_10_	−0.3900 ± 0.1742	−2.24	0.0367
Water temperature/ (°C)	−0.0015 ± 0.0099	−0.15	0.8829
Chl *a*/Log_10_ (µg L^−1^)	0.3144 ± 0.1371	2.29	0.0328
Virus × water temperature	−0.0298 ± 0.0397	−0.75	0.4612
Water temperature × Chl *a*	0.0693 ± 0.0343	2.02	0.0573
Chl *a* × Virus	−0.013019 ± 0.5258	−2.48	0.0224

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
