# Peer review of "Low Host Abundance and High Temperature Determine Switching from Lytic to Lysogenic Cycles in Planktonic Microbial Communities in a Tropical Sea (Red Sea)"

_viruses, 2020, doi:10.3390/v12070761_

Round 1

Reviewer 1 Report

The revisions to the manuscript by Ashy and Agusti have not addressed my previous concerns, and have also added new concerns. There are many reasons why this manuscript should be rejected (including poor writing, data presentation, statistical analysis, and evaluation of previous literature – see below), but the main reason is that the authors appear to be trying to fit the data to a preconceived narrative without sufficient evidence.  I strongly urge the authors to evaluate the data alone, and *then* interpret it.  The authors are missing some additional data that could have helped them interpret the present data (e.g., heterotrophic bacterial production) – that is something that happens in many studies, but the solution should be to address that issue in the manuscript, not to jump to conclusions with insufficient data support.  The manuscript title, abstract, introduction, and discussion still focus on the relationship between types of viral infection and temperature, despite there being no evidence to support this assertion.  The data in this manuscript should be published eventually because it is information, but the manuscript requires considerable reevaluation and reinterpretation to be ready for publication.

Other issues (as mentioned above):

References are still not used accurately in this manuscript.  The authors need to read all cited papers again and use them correctly.  Here are some examples just on the first 2 pages:

Line 44 – Reference 10 does not mention UVB radiation in the context of lysogeny

Lines 50-51 – virus-to-microbe ratios are not related to lysogeny in these references.  Furthermore, VMR is related to viral decay and cell death as much as viral production and bacterial production. Additionally, please see other papers in which lysogeny is definitely not dominant in eutrophic waters (i.e., Boras et al, 2009).

Lines 62-64 – lysogeny is not related to water temperature in these studies. It is related to the lack of organic matter substrate fueling heterotrophic bacterial production.

Lines 70-71 – Bacteriophage 9A is not infective at 30 C because it infects a bacterium in ice brines at < -1 C.  Thus, 30 C and above are definitely not “environmentally-relevant temperatures”.

Lines 72-73 – viral abundance and temperature are related in this study because increased temperatures stabilize the mixed layer, resulting in increased primary productivity and secondary bacterial heterotrophic production, all of which results in greater cell and viral concentrations.

There are also several examples of poor writing and presentation of data. Here are examples (but again, not all instances):

Line 45 – these factors are not “creating a lysogenic cycle”

Line 52 – the dynamics of something cannot become increasingly significant

Lines 69 and 72 – starting a sentence with a bracketed reference number

Figures 4.1 and 4.2 – bidirectional error bars are required. The 2 dates with the highest %lysogeny are lacking error bars.  The dates with zero detectable lysogeny have error bars, despite the caption stating that they do not.

Figures 3.1 and 3.2 – correlations and regression analyses should not be done on the same data. Correlations are performed for independent-independent data comparisons; regressions are performed for dependent-independent datasets.  All of the data in this manuscript are independent.

Table 3 – There is a column with the label “Interactions / Units”. This column only includes units twice and also includes the word “Intercept” for some reason. There are also “x” symbols indicating that a multiplication operation was performed for some reason.

Author Response

Dear Reviewer 1,

Thank you.

Sincerely,

Dr. Ruba A. Ashy

Reviewer 2 Report

The Authors have done a good job in their revision of the original manuscript, addressing the comments raised. I now endorse publication.

Author Response

Dear Reviewer 2,

Thank you.

Sincerely,

Dr. Ruba A. Ashy

Reviewer 3 Report

Dear authors,

this study addresses the dynamic of lytic and lysogenic viruses, as well as heterotrophic bacteria and phytoplankton in a coastal lagoon system on the shores of the Red Sea. The study shows that lytic infection are more prevalent than lysogenic in this system, and that lysogeny is more important when host abundances decline. The study uses well established methods and procedures and is in general very well written. It would have been better to provide further evidence of viral infection as for example including transmission electron microscopy analyses to visualize infected cells. Moreover, it is basically a correlative study and thus the interpretation that during late summer the decline of bacteria is a result of lysogenic virus is not strongly supported. Other reasons (lack of a nutrient, UV, high temperature) could have led to the decline of host cells, and this could have in turn triggered lysogenic virus that initiate a lytic cycle. Finally, flow cytometry should have enabled to identify several populations of viruses (whether if this was the case or not should be mentioned in the text). Such population have been shown to correlate with different host types (see Mojica et al. 2015). Working out this information should improve our understanding of viral diversity and possibly find populations associated with bacteria that would support the results of this study.

Finally, I think you should avoid starting sentences with a reference. Such case occurs throughout the text and should be revised (example: line 69)

Author Response

Dear Reviewer 3,

Thank you.

Sincerely,

Dr. Ruba A. Ashy
